# ALMOST SURE LAST ITERATE CONVERGENCE OF SHARPNESS-AWARE MINIMIZATION

**Kyunghun Nam**
Chung-Ang University
baptist0214@gmail.com

**Jinseok Chung**
POSTECH
jinseokchung@postech.ac.kr

**Namhoon Lee**
POSTECH
namhoonlee@postech.ac.kr

## ABSTRACT

Sharpness-Aware Minimization (SAM) is an iterative optimization process to train neural networks, by which the training is guided to find flat minima, such that the solution found at convergence may generalize well. However, previous studies on the convergence of SAM have only shown the existence of such a solution at arbitrary iteration. We prove that SAM converges at its last iteration almost surely.

## 1 OVERVIEW

Consider the following general optimization problem:

$$\min_{\mathbf{x}} f(\mathbf{x}) \tag{1}$$

where $f$ is the objective function to minimize, convex or non-convex, and $\mathbf{x}$ is its parameters, such as the weights of a neural network in deep learning, for example. The standard approach to (1) is by employing an iterative first-order optimization algorithm such as stochastic gradient descent, which updates $\mathbf{x}$ at each iteration $t$ until it reaches a minimum point $\mathbf{x}^\star$; in non-convex settings, it means finding a stationary point as measured by the gradient norm being zero, $i.e.$, $\|\nabla f(\mathbf{x}^\star)\| = 0$.

Inspired by recent findings in the literature (Yiding et al., 2019; Keskar et al., 2017; Gintare & Roy, 2017) that the shape of minimum highly correlates with their generalization performance, $i.e.$ the flatter it is shaped, the better it is generalized, Foret et al. (2021) have turned (1) into a min-max problem of the following form:

$$\min_{\mathbf{x}} \max_{\|\epsilon\|_2 \leq \rho} f(\mathbf{x} + \epsilon) \tag{2}$$

where $\epsilon$ denotes some perturbation added to $\mathbf{x}$ and $\rho$ sets the radius of the perturbation in 2-norm; $i.e.$, it seeks $\mathbf{x}$ that yields the minimum $f$ in the entire $\epsilon$-neighborhood, which would illustrate "flat" minima in the objective geometry, and hence the name, sharpness-aware minimization or SAM[1]. Specifically, Foret et al. (2021) suggest the following two-step update rule to solve (2) iteratively:

$$\mathbf{x}_{t+\frac{1}{2}} = \mathbf{x}_t + \rho \nabla f(\mathbf{x}_t)/\|\nabla f(\mathbf{x}_t)\|$$
$$\mathbf{x}_{t+1} = \mathbf{x}_t - \eta_t \nabla f(\mathbf{x}_{t+\frac{1}{2}})$$

where $\eta_t$ is the step size. The $\nabla f(\mathbf{x}_t)$ can be replaced with a stochastic version of it in practice and we will prove this setting.

SAM has tremendously succeeded in many scenarios (Chen et al., 2022; Bahri et al., 2022; Na et al., 2022). However, the theoretical understanding of SAM, particularly regarding its convergence behavior, remains quite limited. For example, to our knowledge, the existing convergence results by Andriushchenko & Flammarion (2022); Mi et al. (2022) only show the convergence of SAM at

---

[1]We elaborate more about SAM in Appendix A.

arbitrary iteration, which does not coincide with the current practice of machine learning, *i.e.*, taking the solution achieved at the last iteration[2].

In this work, we reduce this gap by proving almost sure last iterate convergence of SAM. Our analysis is based on the recent work of Liu & Yuan (2022) that proves the same for a family of stochastic gradient methods. Our result shows that SAM can indeed converge to the point of zero gradient norm at convergence under certain conditions.

## 2 LAST ITERATE CONVERGENCE OF SAM

Let $g_t$ be the stochastic gradient of $f$ at $\mathbf{x}_t$. We assume that $g_t$ is $\mathcal{F}_{t+1}$-measurable, meaning its value depends only on the information that is available until that step, but not on any other information that becomes available in the future. The resulting stochastic process $\{\mathbf{x}_t\}$ is adapted to filtration $\mathcal{F}_{t\geq 0}$, which means that it contains all the information of $\mathbf{x}_0, \cdots, \mathbf{x}_t$. To prove the last iterate convergence, we first make the following assumptions[3]:

**(A.1)** There exists $G \geq 0$ *s.t.* $\|\nabla f(\mathbf{x})\| \leq G$ for all $\mathbf{x}$.

**(A.2)** $f$ is L-smooth, *i.e.* $\|\nabla f(\mathbf{x}) - \nabla f(\mathbf{y})\| \leq L\|\mathbf{x} - \mathbf{y}\|$ for all $\{\mathbf{x}, \mathbf{y}\}$.

**(A.3)** $f$ is bounded from below, *i.e.* there exists $f^*$ such that $f(\mathbf{x}) \geq f^*$ for all $\mathbf{x}$.

**(A.4)** $g_t$ is an unbiased estimator of $\nabla f(\mathbf{x}_t)$, and there exist $A, B, C \geq 0$ such that

$$\mathbb{E}[\|g_t - \nabla f(\mathbf{x}_t)\|^2] \leq A(f(\mathbf{x}_t) - f^*) + B\|\nabla f(\mathbf{x}_t)\|^2 + C$$

**(A.5)** The step sizes $\{\eta_t\}$ satisfy $\sum_{t=0}^{\infty} \eta_t = \infty$ and $\sum_{t=0}^{\infty} \eta_t^2 < \infty$ and the perturbation size $\{\rho_t\}$ satisfy $\sum_{t=0}^{\infty} \rho_t^2 < \infty$.

Next, we rely on the following lemma from Orabona (2020); Liu & Yuan (2022):

**Lemma 1.** *Let $\{b_t\}$ and $\{\eta_t\}$ be two nonnegative sequences and $\{\alpha_t\}$ a sequence of vectors. Let $p \geq 1$ and assume $\sum_{t=1}^{\infty} \eta_t b_t^p < \infty$ and $\sum_{t=1}^{\infty} \eta_t = \infty$. Assume also that there exists some $L > 0$ such that $|b_{t+\tau} - b_t| \leq L(\sum_{i=t}^{t+\tau-1} \eta_i b_i + \|\sum_{i=t}^{t+\tau-1} \eta_i \alpha_i\|)$, where $\alpha_i$ is such that $\|\sum_{t=1}^{\infty} \eta_t \alpha_t\| < \infty$. Then $b_t$ converges to 0.*

We are now prepared to present the theorem, which can be formally stated as follows:

**Theorem 1.** *Consider the iterates of SAM under Assumptions $(A.1) \sim (A.5)$ and Lemma 1, then the gradient norm approach zero almost surely, i.e,*

$$\lim_{t\to\infty} \|\nabla f(\mathbf{x}_t)\| = 0 \quad almost\ surely.$$

We direct the reader to Appendix E for the complete proof.

## 3 CONCLUSION

We show that the last iterate of SAM converges almost surely under the standard assumptions, which supports the common practice of using the last iterate as a solution to (2). We note however that this result does not imply how fast it converges or that SAM converges to "flat" minima. We plan to investigate further into these aspects in future work.

## 4 URM STATEMENT

The authors acknowledge that at least one key author of this work meets the URM criteria of ICLR 2023 Tiny Papers Track.

---

[2]We explain more about the difference between the average and last iterate convergences in Appendix B.
[3]We note that these assumptions are considered relatively weak and frequently used in the literature; we further refer to Appendix C for detailed explanations about these assumptions.

## 5 ACKNOWLEDGEMENT

This work was supported by Institute of Information  communications Technology Planning  Evaluation (IITP) grant funded by the Korea government (MSIT) (No.2019-0-01906, Artificial Intelligence Graduate School Program (POSTECH) and No.2022-0-00959, (part2) Few-Shot learning of Causal Inference in Vision and Language for Decision Making) and National Research Foundation of Korea (NRF) grant funded by the Korea government (MSIT) (NRF-2022R1F1A1064569, RS-2023-00210466)

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

## A  Sharpness-Aware Minimization

Deep neural networks are often overparameterized and capable of memorizing the entire training set (Zhang et al., 2017) and it means neural networks can easily achieve extremely small training loss. However, overfitting to the training dataset doesn't mean good generalization ability (e.g (Zhang et al., 2017)), and the training loss landscape of such a complex model has many local and global minima of different generalization abilities. Meanwhile, the "flat" minima (Hochreiter & Schmidhuber, 1997) started to focus again and several studies show that the flatness of the minima is linked with better model generalization (Yiding et al., 2019; Keskar et al., 2017; Gintare & Roy, 2017). For minimizing the sharpness to improve generalization, Foret et al. (2021) proposes a SAM optimizer for converging to flat minima. The sharpness-aware optimization problem is defined as

$$\min_{\mathbf{x}} \max_{\|\epsilon\|_2 \leq \rho} f(\mathbf{x} + \epsilon)$$

To show explicitly that SAM simultaneously minimizes train loss and loss sharpness, we can rewrite the SAM optimization problem as follow.

$$\min_{\mathbf{x}} [\max_{\|\epsilon\|_2 \leq \rho} f(\mathbf{x} + \epsilon) - f(\mathbf{x})] + f(\mathbf{x})$$

SAM optimization problem minimizes the sharpness ($\max_{\|\epsilon\|_2 \leq \rho} f(\mathbf{x} + \epsilon) - f(\mathbf{x})$) and the train loss $f(\mathbf{x})$. In order to minimize the SAM loss function by SGD and its variants, Foret et al. (2021) performs two rounds of approximation. The first approximation is used in calculating the $\epsilon$ at each step. Since the exact solution of the inner maximization $\epsilon^* := \arg\max_{\|\epsilon\|_2 \leq \rho} f(\mathbf{x} + \epsilon)$ is NP-hard problem, they employ an first-order approximation:

$$\hat{\epsilon}(\mathbf{x}) := \arg\max_{\|\epsilon\|_2 \leq \rho_t} f(\mathbf{x}) + \epsilon^T \nabla f(\mathbf{x}) = \rho \frac{\nabla f(\mathbf{x})}{\|\nabla f(\mathbf{x})\|}$$

where $\hat{\epsilon}(\mathbf{x})$ is approximated version of $\epsilon^*$. The second approximation is used to calculate the SAM loss function gradient, i.e. $\nabla f(\mathbf{x} + \hat{\epsilon})$.

$$\nabla f(\mathbf{x} + \hat{\epsilon}) = \frac{d(\mathbf{x} + \hat{\epsilon}(\mathbf{x}))}{d\mathbf{x}} \nabla f(\mathbf{x})|_{\mathbf{x}+\hat{\epsilon}(\mathbf{x})}$$

$$= \underbrace{\nabla f(\mathbf{x})|_{\mathbf{x}+\hat{\epsilon}(\mathbf{x})}}_{a_t} + \underbrace{\frac{d(\hat{\epsilon}(\mathbf{x}))}{d\mathbf{x}} \nabla f(\mathbf{x})|_{\mathbf{x}+\hat{\epsilon}(\mathbf{x})}}_{b_t}$$

Since the $\hat{\epsilon}(\mathbf{x})$ contains $\nabla f(\mathbf{x})$, the computation for $b_t$ needs the Hessian-vector product. For efficient computation, they drop the $b_t$ and the final SAM gradient is derived.

$$\nabla f(\mathbf{x})|_{\mathbf{x}+\hat{\epsilon}(\mathbf{x})}$$

As a result, we can write the iteration of SAM:
$$\begin{cases} \mathbf{x}_{t+\frac{1}{2}} = \mathbf{x}_t + \rho \dfrac{\nabla f(\mathbf{x}_t)}{\|\nabla f(\mathbf{x}_t)\|} \\ \mathbf{x}_{t+1} = \mathbf{x}_t - \eta_t \nabla f(\mathbf{x}_{t+\frac{1}{2}}) \end{cases}$$

## B  Convergence of time average vs Last iterate convergence

In this subsection, we provide additional information to explain the difference between time average convergence and almost surely last iterate convergence results.

When the function $f$ is $L$-smooth, i.e. $\|\nabla f(\mathbf{x}) - \nabla f(\mathbf{y})\|_2 \leq L\|\mathbf{x} - \mathbf{y}\|_2$, approaching a local minimum causes the gradient to go to 0. Therefore, reducing the norm of the gradient is our objective. The vast majority type for the non-convex convergence analysis which is called the convergence of time average usually has the following form:

$$\lim_{T \to \infty} \mathbb{E}\|\nabla f(\mathbf{x}_i)\|^2 = 0 \text{ or } \lim_{T \to \infty} \frac{1}{T} \sum_{t=1}^{T} \mathbb{E}\|\nabla f(\mathbf{x}_t)\|^2 = 0 \qquad (3)$$

where $\mathbf{x}_i$ is an one iterate of optimizer uniformly at random among $\{\mathbf{x}_1, \cdots, \mathbf{x}_T\}$. Another type is called best iterate convergence has the following form:

$$\lim_{T \to \infty} \min_{t \in [T]} \mathbb{E} \|\nabla f(\mathbf{x}_t)\|^2 = 0 \tag{4}$$

(3) or (4) suggests that running an optimizer for $T$ finite iterations, then stopping and returning one of the T iterations at random, yields a small expected norm. However, this only guarantees the existence of at least one $\mathbf{x}_t$ with a small expected norm, but we don't know which one. Hence, (3) or (4) only partially justifies that the last iterate is a good solution ($i.e$ $\|\nabla f(\mathbf{x}_T)\| \approx 0$ for sufficiently large $T$) as a model output.

The last iterate convergence has the following form:

$$\lim_{t \to \infty} \mathbb{E} \|\nabla f(\mathbf{x}_t)\|^2 = 0 \tag{5}$$

or

$$\lim_{t \to \infty} \|\nabla f(\mathbf{x}_t)\|^2 = 0 \tag{6}$$

The result (5) describes the optimizer algorithm by averaging infinitely many runs, but in practice, the algorithm is usually run only once, and the last iterate is returned as the solution. (6) shows that for sufficiently large $T$, the norm of the gradient at the last iterate (or during the end phase of the training) is very small ($i.e$ as $[\|\nabla f(\mathbf{x}_t)\|]_{t \geq 0}$ converges to zero, the term $[\|\nabla f(\mathbf{x}_T)\|]$ will stay small for all $T$ sufficiently large). Hence, (6) characterizes whether an algorithm can eventually approach an exact stationary point or not.

## C    DISCUSSION ABOUT ASSUMPTIONS

**(A.1)** There exists $G \geq 0$ $s.t.$ $\|\nabla f(\mathbf{x})\| \leq G$ for all $\mathbf{x}$.

**(A.2)** $f$ is L-smooth, $i.e.$ $\|\nabla f(\mathbf{x}) - \nabla f(\mathbf{y})\| \leq L\|\mathbf{x} - \mathbf{y}\|$ for all $\{\mathbf{x}, \mathbf{y}\}$.

**(A.3)** $f$ is bounded from below, $i.e.$ there exists $f^*$ such that $f(\mathbf{x}) \geq f^*$ for all $\mathbf{x}$.

**(A.4)** (Khaled & Richtarik, 2020). The stochastic gradient $g_t$ is an unbiased estimator of $\nabla f(\mathbf{x}_t)$, and there exist $A, B, C \geq 0$ such that

$$\mathbb{E}[\|g_t - \nabla f(\mathbf{x}_t)\|^2] \leq A(f(\mathbf{x}_t) - f^*) + B\|\nabla f(\mathbf{x}_t)\|^2 + C$$

**(A.5)** The step sizes $\{\eta_t\}$ satisfy $\sum_{t=0}^{\infty} \eta_t = \infty$ and $\sum_{t=0}^{\infty} \eta_t^2 < \infty$ and the perturbation size $\{\rho_t\}$ satisfy $\sum_{t=0}^{\infty} \rho_t^2 < \infty$.

A filtered probability space $(\Omega, \mathcal{F}, \{\mathcal{F}_t\}_{t \geq 0}, \mathbb{P})$ is a probability space with a sequence of sub $\sigma$-algebra, which captures the evolution of information over time. Here, $\Omega$ is the sample space, $\mathcal{F}$ is the $\sigma$-algebra of events, $\{\mathcal{F}_t\}_{t \geq 0}$ is a sequence of sub $\sigma$-algebras of $\mathcal{F}$, called the filtration, and $\mathbb{P}$ is a probability measure on $(\Omega, \mathcal{F})$. The assumptions for the $g_t$ is $\mathcal{F}_{t+1}$-measurable means that at each time $t$, the value of $g_t$ can be analyzed using a framework called $\mathcal{F}$-measure. Given the available information up to that point, this framework allows us to quantify the probability of certain outcomes occurring at each time step. The resulting stochastic process, denoted by $\mathbf{x}_t$, is adapted to the filtration $\mathcal{F}_{t \geq 0}$ means that the process is designed to take into account all available information up to the current time step, and not rely on any future information. This makes the process more accurate and reliable for analyzing its evolution over time. This assumption refers to a stochastic process, which is a mathematical model that describes the evolution of a system over time, where the outcome at each time step is stochastic. By this assumption, we can better understand how the process evolves over time, and make more accurate predictions about its future behavior. The condition **(A.4)** includes several assumptions for modeling the second moment of the stochastic gradient and hence the most general assumption (see $e.g$ (Khaled & Richtarik, 2020)). The condition of step size **(A.5)** is known as the Robbins-Monro condition (Robbins & Monro, 1951) and is widely used in the SGD literature. We assume that $\sum_{t=1}^{\infty} \rho_t^2 < \infty$, allowing us to derive the last iterate convergence analysis for the SAM.

## D  SUPER-MARTINGALE CONVERGENCE THEOREM

**Theorem 2.** *(Robbins & Siegmund, 1971) Let $\{\mathbf{y}_k\}_{k\geq 0}$, $\{p_k\}_{k\geq 0}$, and $\{q_k\}_{k\geq 0}$ be sequences of nonnegative integrable random variables adapted to a filtration $\{\mathcal{F}_k\}_{k\geq 0}$, that is $\sigma$-algebras such that $\mathcal{F}_k \subset \mathcal{F}_{k+1}$ for all $k$. Furthermore, let $\{\beta_k\}_{k\geq 0} \subseteq \mathbb{R}_+$ be a given with $\sum_{k=0}^{\infty} \beta_k < \infty$ and assume that we have*

$$\mathbb{E}(\mathbf{y}_{k+1}|\mathcal{F}_k) \leq (1+\beta_k)\mathbf{y}_k - p_k + q_k$$

*for all $k$ and $\sum_{k=0}^{\infty} q_k < \infty$ almost surely.*

*Then it holds that $\{\mathbf{y}_k\}_{k\geq 0}$ almost surely converges to a nonnegative finite random variable $y$ and $\sum_{k=0}^{\infty} p_k < \infty$ almost surely.*

## E  MISSING PROOF OF THEOREM 1

By the smoothness of the function $f$, we obtain

$$f(\mathbf{x}_{t+1}) \leq f(\mathbf{x}_t) - \eta_t \langle \nabla f(\mathbf{x}_t), g_{t+\frac{1}{2}}\rangle + \frac{L\eta^2}{2}\|g_{t+\frac{1}{2}}\|^2$$

$$= f(\mathbf{x}_t) - \frac{L\eta^2}{2}\|\nabla f(\mathbf{x}_t)\|^2 + \frac{L\eta_t^2}{2}\|\nabla f(\mathbf{x}_t) - g_{t+\frac{1}{2}}\|^2 - (1-L\eta_t)\eta_t\langle\nabla f(\mathbf{x}_t), g_{t+\frac{1}{2}}\rangle$$

$$\leq f(\mathbf{x}_t) - \frac{L\eta_t^2}{2}\|\nabla f(\mathbf{x}_t)\|^2 + L\eta_t^2\|\nabla f(\mathbf{x}_t) - g_t\|^2 + L\eta_t^2\|g_t - g_{t+\frac{1}{2}}\|^2$$
$$\quad - (1-L\eta_t)\eta_t\langle\nabla f(\mathbf{x}_t), g_{t+\frac{1}{2}}\rangle$$

$$\leq f(\mathbf{x}_t) - \frac{L\eta_t^2}{2}\|\nabla f(\mathbf{x}_t)\|^2 + L\eta_t^2\|\nabla f(\mathbf{x}_t) - g_t\|^2 + L^3\eta_t^2\|\mathbf{x}_t - \mathbf{x}_{t+\frac{1}{2}}\|^2$$
$$\quad - (1-L\eta_t)\eta_t\langle\nabla f(\mathbf{x}_t), g_{t+\frac{1}{2}}\rangle$$

$$= f(\mathbf{x}_t) - \frac{L\eta_t^2}{2}\|\nabla f(\mathbf{x}_t)\|^2 + L\eta_t^2\|\nabla f(\mathbf{x}_t) - g_t\|^2 + \eta_t^2 L^3\rho_t^2$$
$$\quad - (1-L\eta_t)\eta_t\langle\nabla f(\mathbf{x}_t), g_{t+\frac{1}{2}}\rangle$$

Taking the conditional expectation and using the variance assumption we obtain

$$\mathbb{E}(f(\mathbf{x}_{t+1}) - f^*|\mathcal{F}_t) \leq f(\mathbf{x}_t) - f^* - \frac{L\eta_t^2}{2}\|\nabla f(\mathbf{x}_t)\|^2 + L\eta^2\mathbb{E}\|\nabla f(\mathbf{x}_t) - g_t\|^2 + \eta_t^2 L^3\rho^2$$
$$\quad - (1-L\eta_t)\eta_t\mathbb{E}\langle\nabla f(\mathbf{x}_t), g_{t+\frac{1}{2}}\rangle$$

$$\leq f(\mathbf{x}_t) - f^* - \frac{L\eta_t^2}{2}\|\nabla f(\mathbf{x}_t)\|^2 + L\eta_t^2(A(f(\mathbf{x}_t) - f^*) + B\|\nabla f(\mathbf{x}_t)\|^2 + C)$$
$$\quad + \eta_t^2 L^3\rho_t^2 - (1-L\eta_t)\eta_t\mathbb{E}\langle\nabla f(\mathbf{x}_t), g_{t+\frac{1}{2}}\rangle$$

$$\leq f(\mathbf{x}_t) - f^* - \frac{L\eta_t^2}{2}\|\nabla f(\mathbf{x}_t)\|^2 + L\eta_t^2(A(f(\mathbf{x}_t) - f^*) + B\|\nabla f(\mathbf{x}_t)\|^2 + C)$$
$$\quad + \eta_t^2 L^3\rho_t^2 - (1-L\eta_t)\eta_t(\frac{1}{2}\|\nabla f(\mathbf{x}_t)\|^2 - L^2\rho_t^2 - L\rho_t\|\nabla f(\mathbf{x}_t)\|)$$

$$= (1+AL\eta_t^2)(f(\mathbf{x}_t) - f^*) - \frac{\eta_t}{2}\|\nabla f(\mathbf{x}_t)\|^2 + LB\eta_t^2\|\nabla f(\mathbf{x}_t)\|^2 + L\eta_t^2 C$$
$$\quad + \eta_t^2 L^3\rho_t^2 + (1-L\eta_t)\eta_t L^2\rho_t^2 + (1-L\eta_t)L\rho_t\|\nabla f(\mathbf{x}_t)\|$$

$$\leq (1+AL\eta_t^2)(f(\mathbf{x}_t) - f^*) - \frac{\eta_t}{4}\|\nabla f(\mathbf{x}_t)\|^2 + L\eta_t^2 C + \eta_t^2 L^3\rho_t^2$$
$$\quad + (1-L\eta_t)\eta_t L^2\rho^2 + (1-L\eta_t)L\rho\|\nabla f(\mathbf{x}_t)\|$$

$$\leq (1+AL\eta_t^2)(f(\mathbf{x}_t) - f^*) - \frac{\eta_t}{4}\|\nabla f(\mathbf{x}_t)\|^2 + L\eta_t^2 C + \eta_t^2 L^3\rho_t^2$$
$$\quad + (1-L\eta_t)\eta_t L^2\rho^2 + (1-L\eta_t)L\rho_t G$$

The third inequality comes from Lemma 2 in Mi et al. (2022). We obtain just before the last inequality provided that $LB\eta_t \leq \frac{1}{4}$. By the supermartingale theorem, $\sum_{t=1}^{\infty}\eta_t\|\nabla f(x_t)\|^2 < \infty$.

Next, we provide complete proof of

$$\sum_{t=1}^{\infty}(L\eta_t^2 C + \eta_t^2 L^3 \rho_t^2 + (1 - L\eta_t)\eta_t L^2 \rho_t^2 + (1 - L\eta_t)\eta_t L\rho_t G) < \infty \qquad (7)$$

**Lemma 2.** *Let $\{\eta_t\}$ and $\{\rho_t\}$ be two non-negative sequences and hold assumption (A.5). Then,*

$$\sum_{t=1}^{\infty} \eta_t \rho_t < \infty$$

*Proof.* Since $\rho_t$ and $\eta_t$ are non-negative and square summable sequences, it holds that

$$\sum_{t=1}^{k} \rho_t^2 \le \sum_{t=1}^{\infty} \rho_t^2 < \infty, \quad \sum_{t=1}^{k} \eta_t^2 \le \sum_{t=1}^{\infty} \eta_t^2 < \infty$$

Then we can get the following

$$\sum_{t=1}^{k} \rho_t \eta_t \le \sqrt{\sum_{t=1}^{k} \rho_t^2} \sqrt{\sum_{t=1}^{k} \eta_t^2} < \infty$$

By monotone convergence theorem, $\sum_{t=1}^{\infty} \eta_t \rho_t < \infty$. $\qquad\square$

**Corollary 2.1.** *Let $\{\eta_t\}$ and $\{\rho_t\}$ be two nonnegative sequences and hold assumption (A.5). Then,*

$$\sum_{t=1}^{\infty} \eta_t \rho_t^2 < \infty$$

The proof of Corollary 2.1 follows easily from Lemma 2 and the inequality (7) follows from Lemma 2 and Corollary 2.1.

Next, We can derive $\alpha_t = g_{t+\frac{1}{2}} - \nabla f(\mathbf{x}_t)$ by following.

$$|\|\nabla f(\mathbf{x}_{t+\tau})\| - \|\nabla f(\mathbf{x}_t)\|| \le \|\nabla f(\mathbf{x}_{t+\tau}) - \nabla f(\mathbf{x}_t)\|$$
$$\le L\|\mathbf{x}_{t+\tau} - \mathbf{x}_t\| = L\|\sum_{i=t}^{t+\tau-1} \eta_i g_{i+\frac{1}{2}}\|$$
$$= L\|\sum_{i=t}^{t+\tau-1} \eta_i \nabla f(\mathbf{x}_i) + \eta_i(g_{i+\frac{1}{2}} - \nabla f(\mathbf{x}_i))\|$$
$$\le L\left(\sum_{i=t}^{t+\tau-1} \eta_i\|\nabla f(\mathbf{x}_i)\| + \|\sum_{i=t}^{t+\tau-1} \eta_i \alpha_i\|\right)$$

Lastly, we need to show that $\|\sum_{t\ge 1} \eta_t \alpha_t\| < \infty$. For this, we can rewrite $\eta_t \alpha_t$ as follows.

$$\eta_t \alpha_t = \eta_t(g_{t+\frac{1}{2}} - \nabla f(\mathbf{x}_{t+\frac{1}{2}})) + \eta_t(\nabla f(\mathbf{x}_{t+\frac{1}{2}}) - \nabla f(\mathbf{x}_t))$$

Step 1 : $M_t = \sum_{i=1}^{t} \eta_i(g_{i+\frac{1}{2}} - \nabla f(\mathbf{x}_{i+\frac{1}{2}}))$ is a martingale bounded and hence converges almost surely (Williams, 1991).

Step 2 : $N_t = \sum_{i=1}^{t} \eta_i(\nabla f(\mathbf{x}_{i+\frac{1}{2}}) - \nabla f(\mathbf{x}_i))$ converges almost surely.

**Proof of Step 1**

It is well known that $M_t$ is martingale bounded if and only if

$$\sum_{t=1}^{\infty} \mathbb{E}[\|M_t - M_{t-1}\|^2] < \infty.$$

So it is verified by

$$\sum_{t=1}^{\infty} \mathbb{E}[\|M_t - M_{t-1}\|^2] \leq \sum_{t=1}^{\infty} \eta_t^2 (A(f(\mathbf{x}_t) - f^*) + B\|\nabla f(\mathbf{x}_t)\|^2 + C)$$

We already know that $f(x_t) - f^*$ converges almost surely, and $\sum_{t=1}^{\infty} \eta_t^2 < \infty$, $\sum_{t=1}^{\infty} \eta_t \|\nabla f(\mathbf{x}_t)\|^2 < \infty$, we can conclude that $\sum_{t=1}^{\infty} \mathbb{E}[\|M_t - M_{t-1}\|^2] < \infty$.

**Proof of Step 2**

By L-smoothness of $f$, we have

$$\sum_{i=1}^{t} \|\eta_i (\nabla f(\mathbf{x}_{i+\frac{1}{2}}) - \nabla f(\mathbf{x}_i))\| \leq \sum_{i=1}^{t} \eta_i L \|\mathbf{x}_{i+\frac{1}{2}} - \mathbf{x}_i\| = \sum_{i=1}^{t} \eta_i L \|\rho_i \frac{g_i}{\|g_i\|}\|$$

$$= L \sum_{i=1}^{t} \eta_i \rho_i$$

$$\leq L \sqrt{\sum_{i=1}^{t} \eta_i^2} \sqrt{\sum_{i=1}^{t} \rho_i^2}$$

$N_t$ converges almost surely is straightforward by the (A.5).

Through step 1 and step 2, $\|\sum_{t=1}^{\infty} \eta_t \alpha_t\|$ converges almost surely. Applying Lemma 1 with $b_t = \|\nabla f(x_t)\|$ and $p = 2$, we can obtain Theorem 1. $\qquad\square$

