# OpenReview forum: "Almost Sure Last Iterate Convergence of Sharpness-Aware Minimization"
_ICLR.cc/2023/TinyPapers — Submitted to Tiny Papers @ ICLR 2023_

### Official Review · Reviewer_tyvR · 2023-03-27

**Confidence:** 4

**Summary Of Contributions:**

Sharpness-aware optimization, where flatness of the minima is encouraged while optimizing any (potentially non-convex) function, is studied w.r.t. convergence of a certain algorithm. In particular, the two-step update rule of Foret et al. (2021) is analyzed, and last iterate is shown to converge a.s.

**Rating:**

High Potential (HP): a submission which meets the reviewing criteria and has potential to make an impact on the field

**Strengths And Weaknesses:**

Strengths:
- Paper is very well-written. Sufficient amount of detail and context is provided to gauge impact and correctness.
- The writing looks well-aware of the contributions and limitations, and presents them upfront.
- The results seem correct as far as I checked.
- All relevant details are in the appendix.

Weaknesses:
- Some minor writing edits would be good (see suggestions).

**Suggested Changes:**

- It is a bit unclear from the write-up where "SAM" is used to refer to the optimization problem or the particular algorithm (two-step update rule of Foret et al., 2021) for it. I recommend reading through various usage of the term to try to make it clearer to the reader to avoid potential confusion/ambiguity in some places.
- Adding 1-2 lines on key insights in proving Theorem 1 would be a nice addition.

Optional minor writing suggestions:
- flatter it shapes --> flatter it is shaped
- typo in footnote 3: iterature --> literature
- in the main body replace "_Proof._" by "_Proof sketch._", and add □ ("tombstone"/"qed") symbol at the end of proofs.

Optional things to think about:
- Does this have connections to adversarially robust training under L2-ball perturbations? It seems flatter minima should be more robust.

---

> ### Author Response · Authors · 2023-05-03
> **Thanks.**
>
> We deeply appreciate your detailed and constructive comments. In response to your feedback, we have revised the expressions in the paper and will upload the revised version to openreview as soon as possible. Once again, thank you for providing valuable insights that greatly helped in improving our paper.

---

### Official Review · Reviewer_A4tP · 2023-03-30

**Confidence:** 4

**Summary Of Contributions:**

This paper studies the convergence of the last iterate of SAM, rather than the time average convergence. The results show that the gradient of the deterministic version of SAM converges to zero as $t$ approaches infinity.

**Rating:**

Great Start (GS): a submission which meets some of the reviewing criteria but has room for improvement

**Strengths And Weaknesses:**

# Strengths
- The paper studies an important question regarding the convergence of the last iterate of SAM.
- The paper provides a discussion of the assumptions made.

# Weaknesses
- The paper only examines the convergence of the deterministic version of SAM and does not provide information on the **convergence rate**.
-  The paper does not investigate the convergence of the stochastic version of SAM, which may limit the practical applicability of the theory.
- The clarity of the paper is acceptable. However, there are some informal expressions used throughout the text. See the suggestions below.
-  While the proofs appear correct, there are some redundant steps on page 6.


**Suggested Changes:**

- The writing could be improved. Some expressions are informal. For example, the sentence "The proof of Corollary 2.1 is straightforward by the Lemma" could be revised to "The proof of Corollary 2.1 follows easily from Lemma 2." The expression "we can get the theorem 1" could be revised to  "we can obtain Theorem 1".
- The intuition behind assumption (A.4) could be discussed with more detail.

---

> ### Author Response · Authors · 2023-05-03
> **Response to reviewr A4tP**
>
> We sincerely appreciate your detailed and constructive feedback. We deeply agree that not providing a convergence rate is an important point for the direction of follow-up research to develop this study further. We also concur with your comments on the inadequacies in writing and expression, and we have addressed these issues in the revised version. However, the last-iterate convergence we proved is for the stochastic version of the SAM algorithm, which is why the paper includes assumptions about the stochastic gradient $g_t$, and this can be verified in the proof process included in the Appendix.
>
> Once again, thank you for providing valuable insights that greatly helped in improving our paper.

---

### Meta-Review · Area_Chair_fwNK · 2023-04-02

**Recommendation:** Invite to present
**Confidence:** 4

**Metareview:**

Pros
- The paper is reasonably clear, correct and all proof details as well as assumptions needed are included (therefore claims are verifiable).

Cons
- As the reviewers note, there are some limits to the usefulness of the setup considered in the paper. It is still a step in the right direction, but comments from the reviewers can be useful for identifying useful future extensions of the work.


**Summary:**

Last iterate convergence of an alternating update algorithm for sharpness-aware minimization is studied and new theoretical results obtained under reasonable assumptions. The details of proofs and assumptions are included, but writing could be improved.

**Comments And Feedback To The Authors:**

Please take the suggestions provided by the reviewers into account for improving the writing and clarity of the paper.

**Reason For Not Giving A Higher Recommendation:**

Mainly writing gaps that can potentially affect clarity in places. The paper is a small but concrete step on improving the understanding of a relevant problem, and has the potential to qualify as "notable" if writing suggestions are taken into account.

**Reason For Not Giving A Lower Recommendation:**

The main weakness of "clarity" is fixable by the authors in my opinion, via a minor but careful revision.

---

### Decision · Program_Chairs · 2023-04-10

Invite to present